# Current Role of Surgery in the Treatment of Neurocysticercosis

**DOI:** 10.3390/pathogens13030218

**Published:** 2024-02-29

**Authors:** Pedro Tadao Hamamoto Filho, Luiz Fernando Norcia, Agnès Fleury, Marco Antônio Zanini

**Affiliations:** 1Botucatu Medical School, UNESP—Universidade Estadual Paulista, Botucatu 18618-686, Brazil; lf.norcia@unesp.br (L.F.N.); marco.a.zanini@unesp.br (M.A.Z.); 2Instituto de Investigaciones Biomedicas, UNAM—Universidad Nacional Autónoma de Mexico, Ciudad de Mexico 14269, Mexico; 3INNN—Instituto Nacional de Neurología y Neurocirugía, Ciudad de Mexico 14269, Mexico

**Keywords:** neurocysticercosis, hydrocephalus, intracranial pressure, inflammation, VP shunt, neuro-endoscopy, epilepsy

## Abstract

Neurocysticercosis (NCC) is a common parasitic disease of the central nervous system (CNS) in low- and middle-income countries. The infection is pleomorphic, caused by the larval form of the cestode, *Taenia solium*, and part of the heterogeneity of its clinical presentations is associated with the localization of the parasite within the CNS. Changes in the current epidemiological trends of NCC indicate that extra-parenchymal NCC is proportionally becoming more frequent. Extraparenchymal NCC is commonly accompanied by raised intracranial hypertension due to hydrocephalus, which is an emergency requiring cyst extirpation by surgical intervention to relieve the symptoms. Although less frequent, parenchymal cysts may also reach giant sizes requiring urgent surgical treatment. Finally, there is an advancement in the comprehension of the association between NCC and epilepsy—and patients with drug-resistant seizures are candidates for surgical treatment. In this narrative review, we summarize the present state of knowledge to update the current trends in the role of surgery in the treatment of NCC.

## 1. Introduction

Neurocysticercosis (NCC) is the most common parasitic disease of the central nervous system (CNS) in many developing countries, including regions of Latin America, sub-Saharan Africa, and southeast Asia [1,2,3]. It is caused by the ingestion of eggs from the adult tapeworm, *Taenia solium*, and is highly associated with poor sanitary conditions. Migratory flows and trips to endemic regions contribute to new cases in Europe and the United States [4,5,6].

After the larvae hatch from the eggs, they enter the systemic circulation and reach the CNS, where they can lodge in the brain parenchyma or in the cerebrospinal fluid (CSF) compartments (brain ventricles or the subarachnoid space). The parenchymal form of the disease (P-NCC) is the most common and it manifests in headaches, seizures, focal neurological deficits, or behavioral changes. P-NCC is highly responsive to medical treatment, and albendazole and praziquantel are commonly used cysticides, with the proven benefit of effectively destroying cysts. Apart from massive infections, P-NCC has a relatively benign course with an achievable cure [7,8]. However, as will be discussed in this review, surgery may be of utility particularly in cases of large cysts that produce important mass effect and in patients with NCC who have drug-resistant epilepsy.

Extraparenchymal NCC (EP-NCC) is associated with higher rates of morbidity and mortality [9,10,11]. In the CSF compartments, it causes meningitis, vasculitis (and, thereby, stroke), and raised intracranial pressure due to hydrocephalus. In addition, EP-NCC is less responsive to medical treatment: the concentration of albendazole sulfoxide varies widely among individuals and the CSF itself is a low-cellular compartment, allowing the parasites to be somehow hidden from the hosts’ immune surveillance [12,13,14].

For medical treatment, two anthelmintics have proved effective since the 1980s: albendazole and praziquantel. Their efficacy is similar, both being significantly more active in parenchymal disease than in extraparenchymal disease. Treatment is administered for 8 to 15 days, and the doses used are generally 15–30 mg/kg/day for albendazole and 50 mg/kg/day for praziquantel. It should be combined with corticoids, mainly in cases of extraparenchymal localization, to avoid inflammatory complications. The current NCC treatment guidelines are based on high-quality evidence-based recommendations for P-NCC. However, for EP-NCC, the strong recommendations are based on low-quality evidence [15]. Therefore, many questions remain relevant to the treatment of EP-NCC. In addition, EP-NCC is associated with conditions that may require surgical treatment, such as raised intracranial pressure (ICP) due to hydrocephalus. Based on the authors’ background, in this review, we analyze the relevance of NCC for both, clinicians and surgeons, the clinical presentations of the disease, and the different surgical modalities that can be employed for the treatment of hydrocephalus, cyst excision, and epilepsy.

## 2. Recent Evidence Regarding Epidemiological Trends in NCC

The heterogeneity of NCC is related to several factors such as the host’s immune system, genetic differences of the parasite across different geographical sites, and the number and localization of the cysts within the CNS [16,17].

Recently, we proposed that differences in infection pressure (the risk of a person ingesting *T. solium* eggs) could play a role in the disease’s heterogeneity as well [18]. Briefly, although it is difficult to assess the exact time between infection and the onset of symptoms, studies investigating NCC among people who travel from non-endemic countries to endemic regions (British soldiers on duty in India) or people from endemic countries who migrated to non-endemic countries (mainly Latins who migrated to United States), have allowed us to estimate that P-NCC has a latent period of 2–5 years, whereas EP-NCC may vary from 10 to 20 years [19,20]. Furthermore, patients with EP-NCC are older than those with P-NCC [11,21,22], which is probably related to a longer preclinical phase of EP-NCC [23]. Accordingly, the ratio of P-NCC cases to EP-NCC is declining in countries with decreasing disease incidence, such as Brazil, Mexico, and South Korea, although in countries with a still high endemicity of NCC, the proportion of EP-NCC is lower [18].

## 3. Pathophysiology of NCC

In the brain parenchyma, the cysts can initially remain undetected by the immune system. This phase corresponds to the vesicular appearance of cysts on brain imaging (clear fluid). The natural evolution of parenchymal cysts is degeneration accompanied by perilesional inflammation. The clear fluid is replaced by a jellylike material and the host organizes a granuloma to isolate the cysts, which finally calcify. This transition may be asymptomatic or may manifest in symptoms—most frequently, seizures—depending on the number and location of the cysts [24]. However, the inflammation is restricted to the brain parenchyma.

The evolution of cysts in the CSF compartments (the brain ventricles and the subarachnoid space) is different. Inflammation may take a longer time to occur because the CSF is a privileged site for the parasite [25,26]. Indeed, the low cellularity of the CSF and the blood-CSF barrier enable the cysts to remain relatively hidden from the host’s immune surveillance. In addition, the large space of the subarachnoid cisterns allows their growth without triggering an immune response. All these factors explain why patients with subarachnoid NCC have a long latent period from infection to the onset of symptoms.

Once the immune system recognizes the cysts’ antigens and activates to destroy the cysts, inflammatory reactions are initiated triggering the symptoms. Inflammation in the CSF compartments may damage the cranial nerves and the adjacent vessels (leading to vasculitis and, therefore, ischemic attacks), and disturb the CSF flow (either by fibrosis in the subarachnoid cisterns or ependymitis within the ventricles), leading to hydrocephalus. In addition, the cysts themselves may obstruct the CSF flow in the narrow spaces of the ventricles and manifest in hydrocephalus even in the absence of inflammation. Finally, hydrocephalus leads to an increase in ICP, with a high risk of death [27,28,29].

Inflammation and mechanical obstruction are the two mechanisms underlying hydrocephalus in NCC. Inflammation within the basal subarachnoid space impairs the normal CSF flow, resulting in the accumulation of CSF in the ventricles [30].

Recent advances in the understanding of the CSF dynamics, including the glymphatic system and a possible new meningeal layer, have questioned the classic understanding of the flow of CSF from the ventricles, passing through the subarachnoid space, to be absorbed in the arachnoid granulations juxtaposed to the venous sinuses [31,32]. Irrespective of the explanations, it is widely established that subarachnoid NCC causes arachnoiditis [28], which, in turn, impairs the CSF dynamics (either by blocking the classically described flow, or, possibly, by changing the vascular permeability to CSF exchanges).

Therefore, surgical intervention in NCC may target the treatment of hydrocephalus or the mechanical excision of cysts. The treatment of NCC-related epilepsy is also noteworthy and will be discussed later.

## 4. Treatment of Hydrocephalus in NCC

Hydrocephalus in NCC can have a chronic or acute presentation. The chronic presentation is less frequent and generally resembles normal pressure hydrocephalus, including apraxic gait, cognitive deficits, and urinary incontinence. In contrast, acute hydrocephalus is a life-threatening condition that requires emergency care [30]. Additionally, treating hydrocephalus needs to be prioritized over cyst management.

Conventionally, hydrocephalus is treated by ventricle-peritoneal (VP) shunts or endoscopic third ventriculostomy. The same options can be used for NCC-associated hydrocephalus.

Studies on VP shunts for hydrocephalus in NCC have highly heterogeneous results. With respect to symptom resolution, rates of success vary from 20 to 90% [33,34], and the need for further surgical procedures varies from 10 to 100% [33,35]. Specifically, regarding shunt malfunction, the reported rates are approximately 50%, with a mean of 1.5 to 3 procedures per patient [36,37]. This variability can be explained by the differences in diagnostic tools used in each study. For example, older studies did not have the availability of magnetic resonance imaging (MRI), and the diagnosis was based on computed tomography (CT), which has a low sensitivity for the detection of EP-NCC [38,39]. A more recent case series was published with better outcomes, and it is possible that advances in neuroimaging played a role in this improvement [40]. Therefore, in the past, simply placing a VP shunt was insufficient to prevent the possible complications associated with the maintenance of viable cysts within the ventricles or subarachnoid space (Figure 1), such as inflammation and obstruction of the ventricles in locations other than the one in which the catheter was placed. This led to some authors advocating the use of postoperative anthelminthic drugs to promote higher longevity of the shunt [37,41]; though this association has not been demonstrated in recent studies [38]. Nonetheless, VP shunts have a high intrinsic risk of infection and malfunction, and the scenario is not different for NCC, especially because in NCC, inflammation within the CSF may persist irrespective of the shunt.

The introduction of endoscopic tools in neurosurgery benefited the management of hydrocephalus and NCC as well. With endoscopy, it is possible to perform the endoscopic third ventriculostomy (ETV, a communication of the third ventricle directly to the subarachnoid space), creating a bypass for the CSF flow, as well as to mechanically remove cysts inside the ventricles (Figure 2) [42,43]. Reports on ETV for hydrocephalus in NCC indicated significant improvement in symptom resolution (70–95%) and non-requirement of further surgical procedures (0–30%) [44,45,46]. In cases with significant inflammatory reactions, there is a risk of ETV failure due to the closure of the ostomy—and a subsequent VP shunt may be required. However, if neuro-endoscopy is used to remove ventricular cysts, third ventriculostomy should be attempted.

Concomitantly, the introduction of volumetric MRI allowed for good visualization of cysts within the CSF compartments [38,47]. Thus, nowadays it is possible to determine the location of the cysts, and whether they can be endoscopically removed at the same operative time as the treatment of hydrocephalus. However, it is important to highlight that endoscopic management is not possible in all hospitals, especially in endemic countries, and therefore, efforts to improve VP shunt management must continue [48]. In addition, more robust evidence comparing VP shunt and neuro-endoscopy for NCC may help policy makers to update the required technologies. A systematic review comparing the techniques is in progress and shall bring useful data (PROSPERO database CRD42021264290).

## 5. Surgical Removal of Cysts

EP-NCC, unlike P-NCC, responds less positively to medical treatment. This fact may be related to the wide individual variation of albendazole metabolites in the CSF, the higher volume of parasites, the nature of the immune response in this compartment, and the use of corticosteroids to prevent complications during anthelminthic treatment [11,12,13,49]. The anthelminthic treatment may trigger the secretion of various antigens from the cysts, eliciting inflammatory responses that determine the symptoms and can be fatal [8,28]. Therefore, corticosteroids are of critical relevance for the treatment of inflammation [50]. However, the control of inflammation is also beneficial to the parasite, because the parasites may use corticosteroid metabolites for their own reproductive process, as shown with *T. crassiceps* (a *T. solium* analogue) in vitro, and probably also because of the immunosuppression associated with their use [51,52,53]. Consequently, many cycles and higher doses of anthelminthics may be necessary to effectively destroy intraventricular and subarachnoid cysts [54].

Another relevant issue is the mass effect that cysts may cause on the adjacent tissues such as cranial nerves, spinal roots, the spinal cord itself, and the narrow points of the CSF flow. The inflammatory reactions that may be triggered upon anthelminthic treatment can either initiate or deteriorate the previous compressive effect. For cranial nerves, there have been reported cases of visual loss, trigeminal pain, and facial nerve dysfunctions secondary to the compressive effects of cisternal cysts [55,56,57,58]. Cysts in the subarachnoid space of the spine can compress the spinal roots and the spinal cord. These cases can present with severe pain or progressive limb palsy [59,60]. Especially in cases of spinal cysts with spinal cord compression, there may be an urgent need for symptom relief and function restoration. In these cases, surgical removal of cysts is sometimes of primary importance, even though recovery may be limited due to inflammatory injury to the spinal cord or arachnoid adhesions [61].

Intraventricular cysts can cause acute or progressive obstruction of CSF flow, leading to hydrocephalus [30]. As discussed earlier, hydrocephalus can be managed either by VP shunt or ETV. Endoscopic surgery has the advantage of being capable of treating hydrocephalus together with removing cysts [62]. Cysts lodged in the third and lateral ventricles can be managed by rigid endoscopes. Even when cysts cannot be directly visualized, continuous irrigation of the ventricle cavities can move cysts to the operative field for further removal (Figure 3) [63]. For cysts in the fourth ventricle, flexible endoscopes are an interesting option, even though they have lower availability and reduced image quality. Recently, rigid endoscopes using irrigation and pressure differences across the ventricles have been used to successfully remove fourth ventricle cysts [64,65]. Nevertheless, posterior fossa craniotomy followed by a telovelar approach is a relatively simple and effective route for fourth ventricle cysts [66] (Figure 4).

Cisternal subarachnoid cysts can also be reached by means of simple pterional craniotomy with Sylvian fissure dissection (Figure 5). However, a key parameter prior to surgery indication is the assessment of adjacent inflammation. Contrast-enhancement of structures adjacent to the cysts is a signal of inflammation, and the surgical manipulation of these structures may cause significant damage to cranial nerves, vessels (with increased risk of postoperative ischemic lesions), and the brain parenchyma itself [66]. Therefore, the indication of surgical removal of subarachnoid cysts should be based on: (1) good preoperative imaging to discard inflammation surrounding the cysts, (2) adequate microsurgical armamentarium: a surgical microscope with depth of view control, micro scissors, microdissectors, and bipolar forceps, and (3) expertise in microsurgical skills and anatomical knowledge [67,68,69]. The lack of adequate preoperative workup and surgical tools and skills may add risk for the surgical treatment. Minimally invasive approaches with mini-craniotomy, stereotactic surgery, and endonasal endoscopic surgery can also be used with good results and few complications [70].

Furthermore, the main advantage of surgical removal of the cysts is that their complete removal (even though it is not always possible) eliminates the need for postoperative cysticidal treatment and, sometimes, the need for permanent ventricle-peritoneal shunts for hydrocephalus [71]. The elimination of anthelminthic treatment has the advantage of reducing the risk of inflammation, and therefore, the need for corticosteroids, which have substantial side effects.

Another frequently mentioned concern in the surgical management of NCC is the risk of severe inflammation throughout the CNS in case of cyst rupture during surgery. Postoperative neurological deterioration has been attributed to ruptured cysts that would have released several antigenic components, causing subtle inflammation, dissemination of disease, or anaphylactic reaction [72,73]. However, there is currently a paucity of evidence supporting this hypothesis, and several authors’ experiences with surgical care of neurocysticercosis report no complications following cyst rupture. Considerable saline irrigation is suggested to avoid possible complications associated with cyst rupture [74,75,76].

Finally, it is relevant to mention that some parenchymal cysts may require surgical removal, particularly giant cysts with significant mass effects leading to raised intracranial pressure [77,78]. Patients with giant cysts may be admitted to emergency departments with signs of rapid neurological deterioration and, in these cases, surgical removal is more advisable than anthelminthic treatment, which could deteriorate symptoms (due to inflammation) and take several days to reduce the cyst size (Figure 6).

## 6. Surgical Treatment of NCC-Related Epilepsy

The association between NCC and epilepsy has been debated for many years [79,80]. It is typical to refer to NCC as the primary cause of late-onset epilepsy in endemic areas. However, more recent data on the etiology of first acute symptomatic seizures have shown that this may not be the case [81], even though, for NCC, seizures are the most common symptom [82]. Nonetheless, the findings of both, epilepsy and NCC, are not uncommon [83] and further research is needed to determine whether the association is casual or causal.

For surgical purposes, two conditions need to be discussed. First, it has been shown that even calcified cysts may have antigenic remnants that can eventually cause inflammation and trigger seizures [28,84,85]. In addition, the surrounding perilesional edema, gliosis, as well as neuronal and axonal damage may also be epileptogenic foci [86,87,88,89]. Generally, these cases are successfully managed medically with anti-epileptic drugs. For refractory cases, surgical indication is challenging and relies on clear electroencephalographic and neuroimaging evidence that the calcified cyst is the causative lesion for anti-epileptic drug resistance [90,91]. Importantly, intraoperative identification of calcified cysts may be difficult, especially if deeply seated. For lesionectomy in such cases, neuronavigation, focused ultrasound, intraoperative MRI, and electrocorticography are important tools for safer resection with better seizure outcomes [92,93,94].

The second scenario is mesial temporal lobe epilepsy and hippocampus sclerosis (Figure 7). Debate persists whether the nature of this association is causal or casual. Many studies have investigated the relationship between NCC and hippocampus sclerosis and there are doubts whether hippocampus sclerosis results from recurrent seizures from local or distant focus or from chronic recurrent inflammation [95,96,97]. For example, status epilepticus from a distant lesion could lead to hippocampus injury; NCC can induce gliosis surrounding the hippocampus causing damage to it; antibodies against NCC could have a cross-reaction against hippocampal cells; the hippocampus being a highly sensitive structure of the brain, can be damaged secondary to systemic conditions (such as febrile seizures) [98,99,100,101]. Several hypotheses exist on the pathophysiologic mechanisms underlying NCC and mesial temporal lobe epilepsy [102]. Irrespective of the precise mechanisms, patients with temporal lobe epilepsy due to hippocampus sclerosis clearly benefit from surgical treatment [103]. The surgical procedure may vary from selective amygdalohippocampectomy (AH) to a combination of AH and anterior temporal lobectomy [104]. Moreover, for AH, there is disagreement regarding transsylvian or transcortical approaches [105]. According to the authors’ experience, there are significant disparities between selective AH and anterior temporal lobectomy in terms of seizure and cognitive results. [106,107,108]. However, for selective AH, there are differences with regard to postoperative visual defects, which are higher for transcortical rather than for transsylvian approaches [109]. Specifically for NCC and hippocampus sclerosis, there is a lack of studies comparing the techniques and approaches. Nonetheless, it appears reasonable that in cases of calcified lesions close but outside the hippocampus, a more extensive resection would be recommended.

## 7. Concluding Remarks

Despite the decrease in NCC globally, it remains a neglected disease with enormous consequences for patients and governments [110,111,112,113,114]. The accurate identification of NCC and its differential diagnosis remains important, especially in non-endemic and decreasing endemicity contexts [115]. In this regard, the role of neurosurgery in disease management remains relevant.

If hydrocephalus is present, endoscopic procedures have the advantage of treating hydrocephalus and removing intraventricular cysts. Attention should be paid to the identification of cysts attached to the ventricle walls; in these circumstances, extensive cyst removal may be detrimental and result in deficits. In addition, even after treatment for cysts (either by surgical removal or anthelminthic drugs) and third ventriculostomy, hydrocephalus may occur at a later stage, and a VP shunt may have to be performed. The surgical removal of the subarachnoid is feasible and may dismiss the use of postoperative anthelminthic drugs if all cysts are removed. However, for subarachnoid cysts surgery, it is mandatory to verify that there are no signals of active inflammation. As for endoscopic procedures, cysts attached to vascular or neural structures should not be removed. Research is needed to clarify the relationship between NCC and epilepsy. Irrespective of the precise pathophysiologic mechanisms, calcified cysts will rarely demand lesionectomy, whereas hippocampus sclerosis causing mesial temporal lobe epilepsy benefits from surgical treatment.

## Figures and Tables

**Figure 1 pathogens-13-00218-f001:**
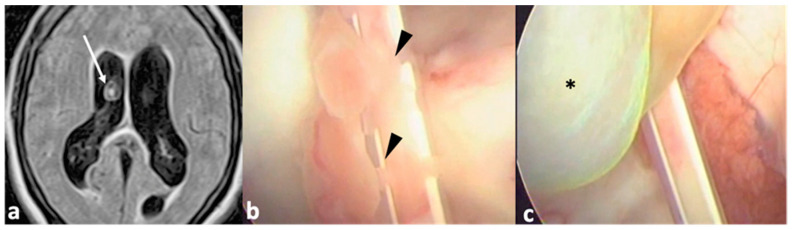
Patient with hydrocephalus secondary to NCC. She was first managed by a VP shunt, which developed a malfunction a few months after insertion. A magnetic resonance imaging (**a**) showed a cyst (arrow) within the right ventricle. A new surgery (**b**) was performed for endoscopic removal of the cyst. During ventricle examination, choroid scars were identified near the previous catheter (arrowheads), which were the cause of the shunt malfunction. Finally, the cyst (**c**), * was successfully removed.

**Figure 2 pathogens-13-00218-f002:**
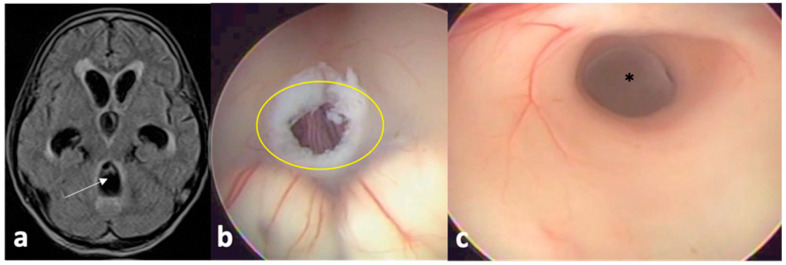
MRI of a patient with hydrocephalus (**a**) secondary to a cysticercus occupying the fourth ventricle towards the Sylvius aqueduct. He underwent a conventional endoscopic third ventriculostomy (**b**) with communication of the third ventricle to the subarachnoid space; after rotating the endoscope, the cyst (*) could be identified within the aqueduct (**c**) with posterior mechanical removal.

**Figure 3 pathogens-13-00218-f003:**
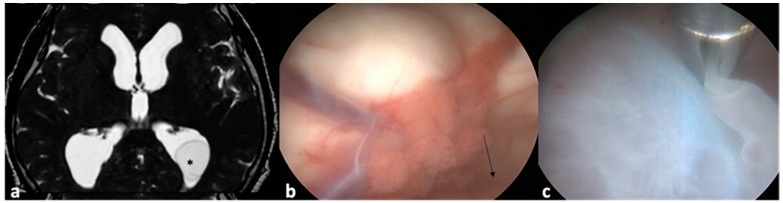
In this MRI, a cyst (*) can be visualized within the occipital horn of the left lateral ventricle (**a**). The endoscope was introduced through a conventional burr-hole in the Kocher point until the lateral ventricle (**b**) and slightly turned posteriorly (arrow) towards the ventricular atrium. After continuous saline irrigation, the cyst moved from the occipital horn to the operative field and could be held with surgical forceps (**c**).

**Figure 4 pathogens-13-00218-f004:**
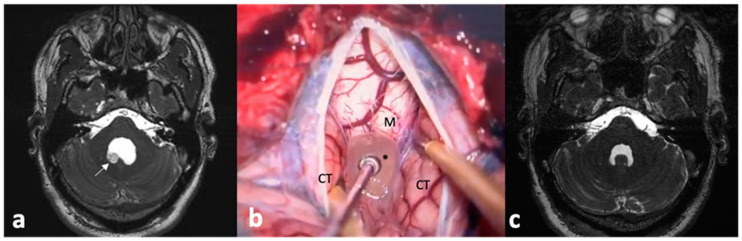
(**a**) This MRI shows a cyst with a scolex (arrow) within the fourth ventricle. This cyst was removed by means of suboccipital craniotomy (**b**) with dissection of the cerebellar tonsils (CT) to expose the fourth ventricle, allowing for the removal of the cyst (*), M: medulla oblongata. Postoperative imaging (**c**) showed no residual lesion and recovery of the normal shape of the fourth ventricle.

**Figure 5 pathogens-13-00218-f005:**
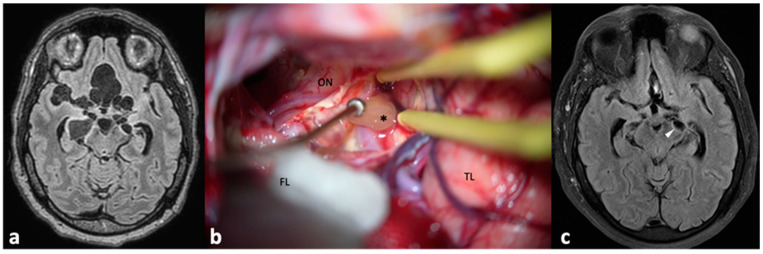
Preoperative magnetic resonance imaging of a patient with multiple subarachnoid cysts (**a**) within several cisterns. Most of the cysts were successfully and uneventfully removed by means of a conventional pterional approach (**b**), ON: optic nerve, FL: frontal lobe, covered by cotton compress, TL: temporal lobe, *: a cyst in the carotid–oculomotor cistern. (**c**) Postoperative imaging showed a residual cyst (arrowhead, contralateral to the approach side) that was treated with albendazole.

**Figure 6 pathogens-13-00218-f006:**
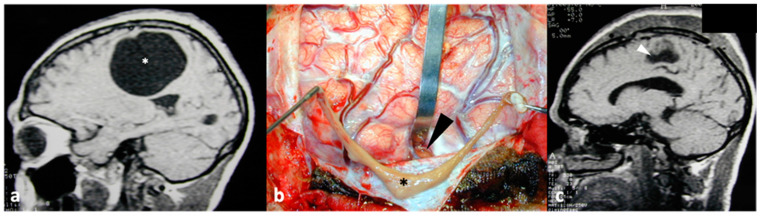
Preoperative magnetic resonance imaging showing a giant parenchymal cysticercus ((**a**), *). The cyst (*) was entirely removed through minimum brain damage with a small corticectomy ((**b**), arrowhead). Postoperative imaging (**c**) showed only an area of encephalomalacia (arrowhead).

**Figure 7 pathogens-13-00218-f007:**
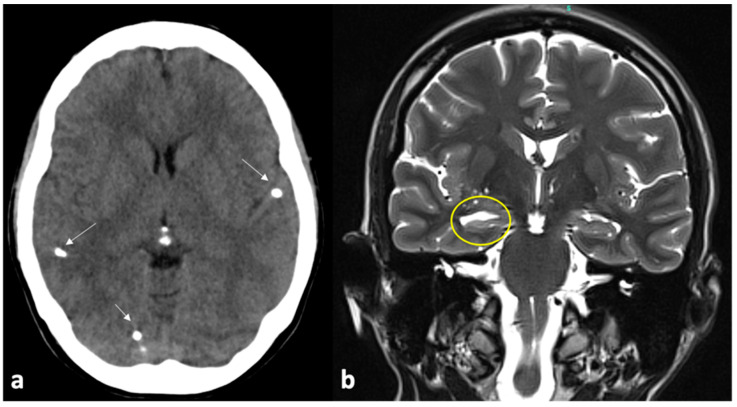
This is the case of a patient with epilepsy who presented with multiple calcified lesions (arrows) on computed tomography (**a**). Magnetic resonance imaging showed sclerosis of the right hippocampus, clearly seen by means of an enlarged temporal horn ((**b**), yellow circle). After anterior temporal lobectomy and amygdalohippocampectomy, the patient had significant improvement in seizures.

## Data Availability

Not applicable.

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
