# Peer review of "Current Role of Surgery in the Treatment of Neurocysticercosis"

_pathogens, 2024, doi:10.3390/pathogens13030218_

Round 1

Reviewer 1 Report

Comments and Suggestions for Authors

General comments

the authors present a summary of their own practice as well as recent publications on the use of surgery in the treatment of NCC. This is a very interesting text .

However the format uses by the authors is a little confusing as they don't follow a usual scheme for systematic review, and don’t. Indeed, they don't present their sources of articles, the keywords used and the flow chart describing their search strategy for publications. We can rapidly understand that the text is more related to their own work than formal review of the literature. Most of the time they refer to their own practice more than other paper.

As this team seems to have one of the largest experience in the field, they should have presented the paper as the summary of xxx years of practice than review of the literature, and discuss it in regard with other publications. Literature review seems not the appropriate format of this text. 

More in detail:

the first paragraph is quite surprising as they claim for a delay for appearance of clinical epilepsy of 2-3 years for P-NCC to 20 years for EP-NCC based of traveler clinical investigations.  We agree that is approach sounds interesting, however a large part of patients attending dispensary are children, with an active infection instead of calcified parasites. As all the other parts of the text describe technical surgical approaches and results, the pathophysiological aspect of the disease should be emphasized.

Author Response

General comments

The authors present a summary of their own practice as well as recent publications on the use of surgery in the treatment of NCC. This is a very interesting text .

Thank you for revising our manuscript.

However the format uses by the authors is a little confusing as they don't follow a usual scheme for systematic review, and don’t. Indeed, they don't present their sources of articles, the keywords used and the flow chart describing their search strategy for publications. We can rapidly understand that the text is more related to their own work than formal review of the literature. Most of the time they refer to their own practice more than other paper.

As this team seems to have one of the largest experience in the field, they should have presented the paper as the summary of xxx years of practice than review of the literature, and discuss it in regard with other publications. Literature review seems not the appropriate format of this text. 

We understand the reviewer's point. Nevertheless, narrative reviews allow for a more flexible approach to the research question, dismissing the traditional structure of systematic reviews. Anyhow, we accept the reviewer' suggestion and we added the information that the text is based on the authors' experience.

More in detail:

The first paragraph is quite surprising as they claim for a delay for appearance of clinical epilepsy of 2-3 years for P-NCC to 20 years for EP-NCC based of traveler clinical investigations.  We agree that is approach sounds interesting, however a large part of patients attending dispensary are children, with an active infection instead of calcified parasites. As all the other parts of the text describe technical surgical approaches and results, the pathophysiological aspect of the disease should be emphasized.

With all due respect, we disagree with the reviewer. NCC is far more common in adults than in children. 
Regarding pathophysiology, it is presented on section 3. A yet deeper approach on the (complex) pathophysiology of NCC is beyond the scope of the present text. Even though NCC can be an important cause of seizures among children in India (as referred by some papers), this scenario is not enough to state that children are more affected than adults.

Reviewer 2 Report

Comments and Suggestions for Authors

I congratulate authors for their manuscript that touches a relatively neglected area in neurology and neurosurgery. Very good review of current surgical treatment options. I recommend adding the specific medical management recommended (dosing, duration). 

Author Response

I congratulate authors for their manuscript that touches a relatively neglected area in neurology and neurosurgery. Very good review of current surgical treatment options. I recommend adding the specific medical management recommended (dosing, duration). 

We thank the reviewer for appreciating our manuscript.
We added a brief of medical treatment of NCC as suggested by the reviewer. However, deeper insights on this topic have been exhaustively published elsewhere and is beyond the scope of this paper, which is targeted surgical treatment.

Round 2

Reviewer 1 Report

Comments and Suggestions for Authors

the authors focussed on their own experience which is the best way to present this paper instead of a systematic review